# A dynamic knowledge graph approach to distributed self-driving laboratories

Jiaru Bai [1], Sebastian Mosbach [1,2], Connor J. Taylor[3,4,8], Dogancan Karan[2], Kok Foong Lee[5], Simon D. Rihm [1,2], Jethro Akroyd [1,2], Alexei A. Lapkin [1,2,4] & Markus Kraft [1,2,6,7] ✉

The ability to integrate resources and share knowledge across organisations empowers scientists to expedite the scientific discovery process. This is especially crucial in addressing emerging global challenges that require global solutions. In this work, we develop an architecture for distributed self-driving laboratories within The World Avatar project, which seeks to create an all-encompassing digital twin based on a dynamic knowledge graph. We employ ontologies to capture data and material flows in design-make-test-analyse cycles, utilising autonomous agents as executable knowledge components to carry out the experimentation workflow. Data provenance is recorded to ensure its findability, accessibility, interoperability, and reusability. We demonstrate the practical application of our framework by linking two robots in Cambridge and Singapore for a collaborative closed-loop optimisation for a pharmaceutically-relevant aldol condensation reaction in real-time. The knowledge graph autonomously evolves toward the scientist's research goals, with the two robots effectively generating a Pareto front for cost-yield optimisation in three days.

The concept of laboratory automation, recently reinterpreted as self-driving laboratories (SDLs)[1,2], has been in existence since the 1960s, when ref. 3 introduced the first automated chemistry hardware. Since then, SDLs have gained widespread adoption in chemistry[4–7], materials science[8,9], biotechnology[10,11] and robotics[12], resulting in accelerated scientific discovery and societal development. However, the implementation of SDLs can be challenging and typically requires a highly specialised team of researchers with expertise in chemistry, engineering, and computer science. Consequently, studies are often conducted by large research groups within a single organisation. Even in cases where collaborations occur between research groups, the SDL is usually centralised within the same laboratory.

In response to the pressing global challenges of today, there is a growing consensus within the scientific community that a paradigm shift towards a globally collaborative research network is necessary[13–15]. This shift requires decentralising SDLs to integrate different research groups to contribute their expertise towards solving emerging problems[16]. Such decentralisation holds great potential in supporting various tasks ranging from automating the characterisation of epistemic uncertainty in experimental research[17] to advancing human exploration in deep space[18]. Achieving this vision is not an easy task and entails three major challenges. The first challenge is efficiently orchestrating heterogeneous resources[19], which includes hardware from different vendors and diverse computing environments.

[1]Department of Chemical Engineering and Biotechnology, University of Cambridge, Philippa Fawcett Drive, Cambridge CB3 0AS, UK. [2]Cambridge Centre for Advanced Research and Education in Singapore (CARES), 1 Create Way, CREATE Tower, #05-05, Singapore 138602, Singapore. [3]Astex Pharmaceuticals, 436 Cambridge Science Park Milton Road, Cambridge CB4 0QA, UK. [4]Innovation Centre in Digital Molecular Technologies, Yusuf Hamied Department of Chemistry, University of Cambridge, Lensfield Road, Cambridge CB2 1EW, UK. [5]CMCL Innovations, Sheraton House, Cambridge CB3 0AX, UK. [6]School of Chemical and Biomedical Engineering, Nanyang Technological University, 62 Nanyang Drive, 637459 Singapore, Singapore. [7]The Alan Turing Institute, London NW1 2DB, UK. [8]Present address: Faculty of Engineering, University of Nottingham, University Park, Nottingham NG7 2RD, UK.
✉e-mail: mk306@cam.ac.uk

The second challenge is sharing data across organisations[20], which requires standardising language in which the research is communicated[21]. During this process, the source and metadata of the research need to be tracked to facilitate reproducibility, which leads to the third challenge of data provenance recording following FAIR principles – Findable, Accessible, Interoperable and Reusable[22].

Many attempts have been made to tackle these challenges with different focuses. For resource orchestration, middleware such as ChemOS[23], ESCALATE[24], and HELAO[25] exist to glue different components within an SDL and abstract the hardware resources. For data sharing, χDL[26,27] and AnIML[28] are examples of standard protocols developed for synthesis and analysis respectively. In the realm of data provenance, Mitchell et al.[29] proposed a data pipeline to support the modelling of the COVID pandemic, whereas ref. 30 devised a knowledge graph to record experiment provenance in materials research. Although these studies provide insights into building a collaborative research environment, they are developed in isolation with customised data interfaces. Enhancing interoperability both within and between these systems is essential to establish a truly connected research network.

As discussed in our previous work[31,32], semantic web technologies such as knowledge graphs[33] offer a viable path forward. Ontologies abstract both resources and data using the same notion, allowing for a common language between participants when allocating tasks and sharing results. The World Avatar[34,35] is such a knowledge graph that aims to encompass all aspects of scientific research laboratories as shown in Fig. 1a in their entirety: The experiment itself, including its physical setup and underlying chemistry; moving handlers that can be of human or robotic nature; and the laboratory providing necessary infrastructure and resources[36]. The World Avatar goes beyond static knowledge representation by encoding software agents as executable knowledge components, enabling dynamicity and continuous incorporation of new concepts and data while preserving connections to existing information. As the knowledge graph expands, this characteristic allows for capturing data provenance from experimental processes as knowledge statements, effectively acting as a living copy of the real world. This dynamic knowledge graph streamlines the immediate dissemination of data between SDLs, offering a promising holistic solution to the aforementioned challenges[32,37] and the pursuit of the Nobel Turing Challenge[36,38].

In this work, we demonstrate a proof-of-concept for a distributed network of SDLs enabled by a dynamic knowledge graph. This signifies the first step towards digital research scientists (as shown in Fig. 1a) collaborating autonomously. To illustrate the effectiveness of this approach, as shown in Fig. 1b, we present a demonstration using two robots in Cambridge and Singapore collaborating on a multi-objective closed-loop optimisation problem in response to a goal request from scientists.

## Results

### Architecture of distributed SDLs

Closed-loop optimisation in SDLs is a dynamic process that revolves around design-make-test-analyse (DMTA) cycles[39,40]. Compared to machine learning systems and scientific workflows that only capture data flows, SDLs offer an integrated approach by orchestrating both computational and physical resources. This involves the integration of data and material flows, as well as the interface that bridges the gap between the virtual and physical worlds. To this end, we propose a conceptual architecture of distributed SDLs that effectively incorporates all three flows, as illustrated in Fig. 2a.

The proposed architecture presents a framework to enable scientists to set research goals and resource restrictions for a particular chemical reaction and have them trigger a closed-loop process in cyberspace. The process is initiated by the monitoring component, which parses the research goals and requests the iterations needed to achieve the objectives. The iterating component collects prior information about the design space and passes it on to the component that designs the next experiment. The algorithm employed, as well as the availability of prior data, determines the combination of design variables to be proposed within the search space provided by the scientist. Subsequently, the proposed physical experimentation is scheduled for execution in one of the available laboratories, similar to the scheduling of high-performance computing jobs[41]. The suggested conditions are translated to the machine-actionable recipe that enables the control of hardware for reaction and characterisation. In the physical world, this is reflected in the material flow between the two pieces of equipment. The data processing component is then responsible for computing the objectives by analysing the complete job information and raw data. If the resources are still available, a comparison of these objectives with the research goals determines whether the system should proceed to the next iteration.

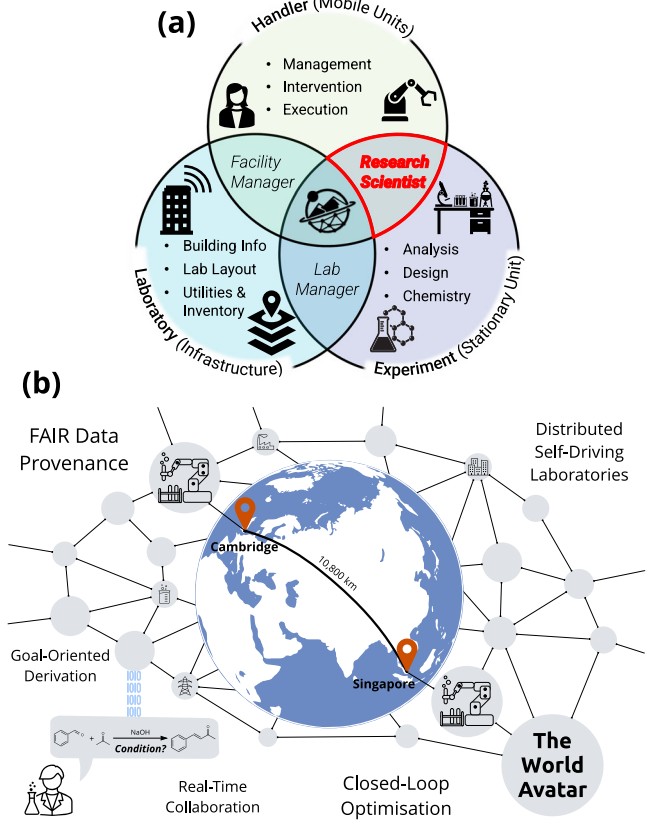

**Fig. 1 | An overview of the World Avatar approach towards globally connected laboratory digital twins. a** Three interrelated aspects of a chemical research laboratory that need to be represented, adapted from[36]. The handler set pertains to the tasks demanding the physical involvement of mobile units. The experiment set includes stationary units, specifically hardware and chemicals. The laboratory set represents the environmental conditions and building infrastructure. The intersecting regions symbolise the nuanced roles within the laboratory, requiring expertise in the delineated sets. At the intersection of these three circles is the World Avatar project, an initiative aiming to proficiently integrate expertise across these essential facets. This paper focuses on the automation of chemical reaction optimisation, a task that can be viewed as part of the daily work of many research scientists. The illustrations of the lab glass and atoms were created using istockphoto.com. **b** Two labs in Cambridge and Singapore are linked to demonstrate real-time collaborative closed-loop optimisation. The process is triggered by a goal request from a research scientist, and all data provenance is preserved. The developed infrastructure in this work contributes to the establishment of distributed self-driving laboratories.

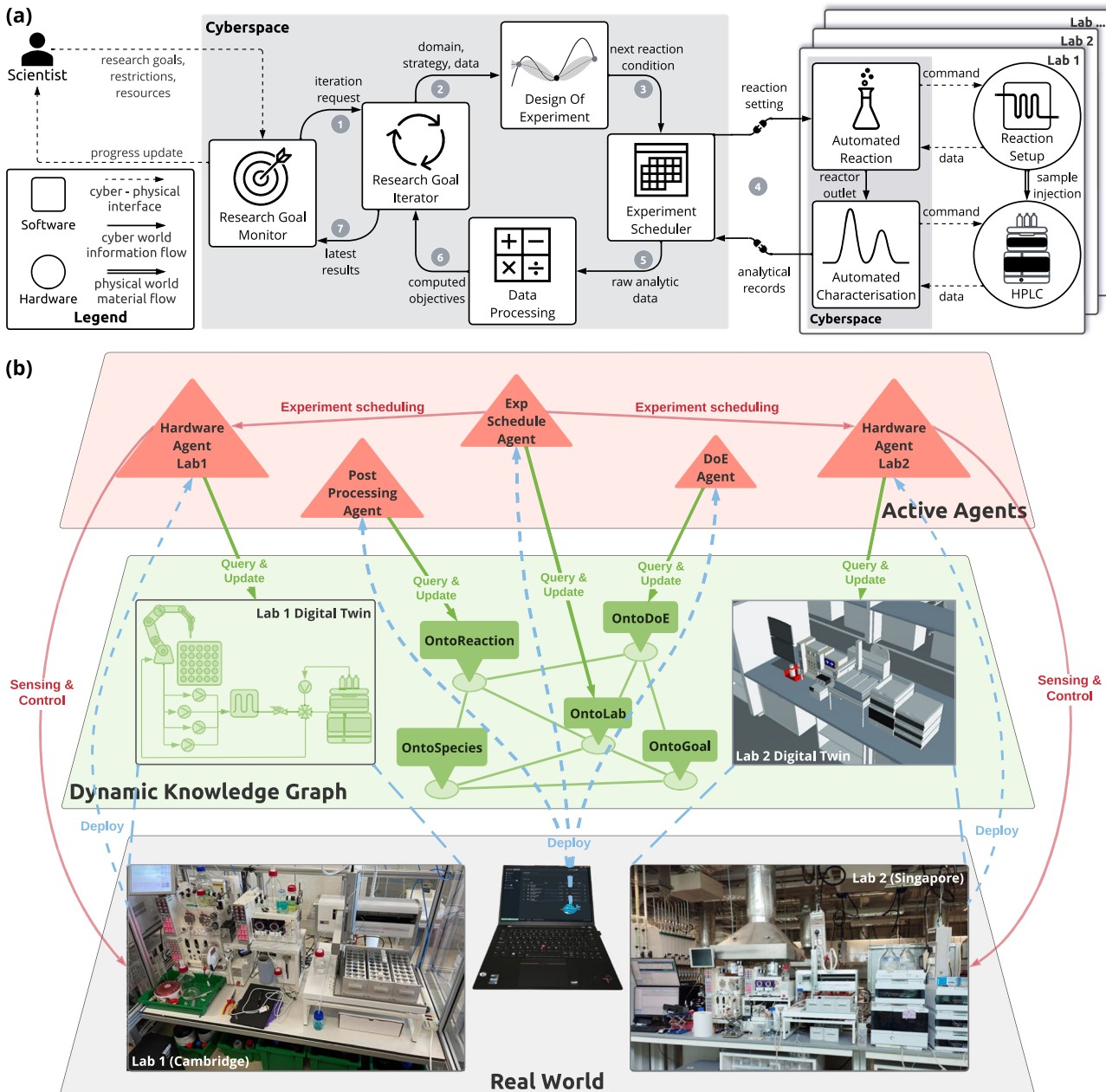

**Fig. 2 | An illustration of a distributed self-driving laboratories (SDLs) architecture. a** Conceptual framework of components used to build a network of distributed SDLs for closed-loop optimisation. The framework encompasses a holistic integration of data, software, hardware, and workflow, taking into account the flow of information within cyberspace and materials within physical space. Initiated by the scientist specifications, these flows autonomously evolve across cyber and physical spaces until they accomplish the research goals or exhaust allocated resources. The illustration of the design of experiments was created by ref. 81.

**b** Dynamic knowledge graph approach that is structured into three layers. The first layer represents the real world, where hardware is located and reactions take place. The second layer consists of a dynamic knowledge graph in cyberspace, hosting information such as the digital twin of the hardware and chemical data. The third layer comprises active agents that continually monitor the status of the knowledge graph, dynamically restructuring it, and actuating changes in the real world. The illustration of docker was created using flickr.com.

This architecture liberates the scientists from routine work, however, it also poses challenges in the implementation in terms of ensuring robustness, scalability, maintainability, safety, and ethics. Ideally, the system should enable seamless integration of new devices, resources, and algorithms without disrupting the system's overall functioning. It is also critical to allow for dynamic adaption to changes in research goals and resource restrictions.

We believe dynamic knowledge graph technology can help with realising this architecture[32]. Specifically, as illustrated in Fig. 2b, this technology abstracts the software components as agents that receive

inputs and produce outputs. The flow of data between these components is represented as messages exchanged among these agents. Physical entities can be virtualised as digital twins in cyberspace, enabling real-time control and eliminating geospatial boundaries when multiple labs are involved. This reformulation of the closed-loop optimisation problem as information travelling through the knowledge graph and reflecting their changes in the real world offers a powerful framework for achieving true distributed SDLs. In this way, we can think of an occurrence of physical experimentation as a sequence of actions that dynamically generates information about a

reaction experiment as it progresses in time, analogous to computational workflows[42].

This work is part of a series of papers introducing a holistic approach to lab automation by including all aspects of research laboratories (see Fig. 1a) in an all-encompassing digital twin[36]. By employing dynamic knowledge graphs that integrate knowledge models from different domains, we can address the challenges related to interoperability and adaptability commonly encountered in platform-based approaches[32]. The goal-driven architecture facilitates reasoning across the knowledge base, allowing high-level, abstract goals to be decomposed into specific sub-goals and more tangible tasks. Within this framework, humans play a dual role, functioning both as goal setters and operators (when necessary) for executing and intervening in experiments. When acting as operators, humans can be represented in the knowledge graph similarly to robots, and they receive instructions in a human-readable format. This facilitates the realisation of a hybrid and evolving digital laboratory, bridging potential "interim technology gaps"[43]. The operations described in this work are carried out through robotic handling, with humans primarily involved in the preparation of initial materials and the maintenance of the equipment.

## Chemical ontologies and digital twins

The realisation of SDLs requires a connection between abstract chemistry knowledge and concrete hardware for execution[21]. This calls for a set of connected ontologies, as identified in our previous analysis on the gaps in current semantic representations for chemical digitalisation[32]. Figure 3 presents a selection of concepts and relationships as an effort to address these gaps. These concepts span various levels of abstraction involved in scientific research, ranging from the high-level research goals, through the conceptual level of chemical reactions and the mathematical level of design of experiments, down to the physical execution of reaction experiments and the laboratory digital twin. We describe below ontologies' cross-domain characteristics, for technical details on each ontology please see Supplementary Information section A.1.

For closed-loop optimisation in SDLs, we draw parallels between the pursuit of optimal objectives and the reasoning cycles involved in pursuing a goal[44,45]. The multi-objective problem can be formulated as a `GoalSet` which comprises individual `Goal`s. Each goal is associated with specified dimensional quantities that can be achieved by a `Plan`, which consists of multiple `Step`s to be carried out by corresponding agents. From the implementation perspective, this is akin to a specialised research sub-domain within the scientific workflow community that focuses on the management of iterative workflows abstracted as directed cyclic graphs[46]. In this regard, we adopt the derived information framework[42], a knowledge-graph-native approach, to manage the iterative workflow.

In developing chemical ontologies for SDLs, we draw upon the lessons learnt in creating ontologies for chemical plants. One prominent example is the OntoCAPE material and chemical process system[47] ontology, which describes materials from three aspects: the `ChemicalSpecies` that reflects the intrinsic characteristics, `Material` as part of the phase system which describes macroscopic thermodynamic behaviour, and `MaterialAmount` that refers to a concrete occurrence of an amount of matter in the physical world. Building on this foundation, we introduce OntoReaction, an ontology that captures knowledge in wet-lab reaction experiments, and OntoDoE, an ontology for the design of experiments (DoE) in optimisation campaigns. As an effort to align with existing data, OntoReaction draws inspiration from established schemas used in chemical reaction databases like ORD[48] and UDM[49]. `ReactionExperiment` is a concrete realisation of a `ChemicalReaction` that is sampled at a set of `ReactionConditions` and measures certain `PerformanceIndicators`. When grouped together, they can form `HistoricalData` that

are utilised by a `DesignOfExperiment` study to propose new experiments.

In the development of our hardware ontologies, we have expanded upon concepts from the Smart Applications REFerence (SAREF) ontology[50], which is widely adopted in the field of the Internet of Things. We introduce OntoLab to represent the digital twin of a laboratory, comprising a group of `LabEquipment` and `ChemicalContainer`s that contain `ChemicalAmount`. Furthermore, we create OntoVapourtec and OntoHPLC as ontologies for the equipment involved in this work, linking them to the concrete realisation aspect of OntoCAPE. We establish the link between abstract chemical knowledge and hardware by translating `ReactionCondition` to `ParameterSetting`, which can be combined to form `EquipmentSettings` for configuration.

## Contextualised reaction informatics

By utilising ontologies as blueprints, we can instantiate reaction information while preserving connections to contextual recordings. The reaction we choose for demonstration is an aldol condensation reaction between benzaldehyde **1** (bold numbers for reference) and acetone **2**, catalysed by sodium hydroxide **3** to yield the target product benzylideneacetone **4**[51], which is pharmaceutically relevant and can be used to treat idiopathic vomiting as an NK-1 receptor inhibitor[52]. Additionally, reported side products include dibenzylideneacetone **5** and further condensation products from acetone polymerisation. The choice of this well-studied reaction is deliberate, aimed at explaining the contribution of our work to developing distributed SDLs to a broad audience, and an application to more interesting chemistry will be presented in a subsequent paper.

Figure 4 provides an illustrative representation of the chosen reaction in the knowledge graph as viewed through various roles within a laboratory, each with its unique perspective on the same chemical. Taking the starting material benzaldehyde as an example, it demonstrates how a knowledge graph can enhance the daily work of different roles. A chemist, more interested in conceptual description, might look at benzaldehyde as a reactant and search for relevant species information. A data scientist might examine its concentration to determine the appropriate usage of other chemicals when designing conditions for a particular reaction experiment. Meanwhile, the 3D digital twin built on top of the knowledge graph offers a lab manager a centralised hub for real-time monitoring of lab status[53], ensuring the availability of an internal standard that can be mixed with the physical existence of benzaldehyde to enable characterisation during the actual execution of the experiment. In practice, the same individual might play several roles, and the emphasis here is on the cross-domain interoperability facilitated by the amalgamation of different aspects into a unified knowledge graph. This integration ensures the relevance of information to a diverse range of users while maintaining human oversight. Consequently, this approach may present opportunities for the enhancement of various digital applications, such as the utilisation of virtual reality for laboratory training[54].

The integration of chemical knowledge from PubChem, represented by OntoSpecies for unique species identification[55], serves as a critical link between these facets of chemicals. It enables the identification of potential input chemicals based on the reactant and solvent during DoE and allows for the selection of appropriate sources of starting materials from multiple chemical containers (see Supplementary Information section A.2). Another aspect enabled by this disambiguation of species relates to the representation of chemical impurities. In this case study, all starting materials were procured and used as received, with purities exceeding 99% for liquid chemicals and 97% for NaOH pellets (see Supplementary Table S4). The impurities are categorised as unknown components, and their presence is indicated using the data property

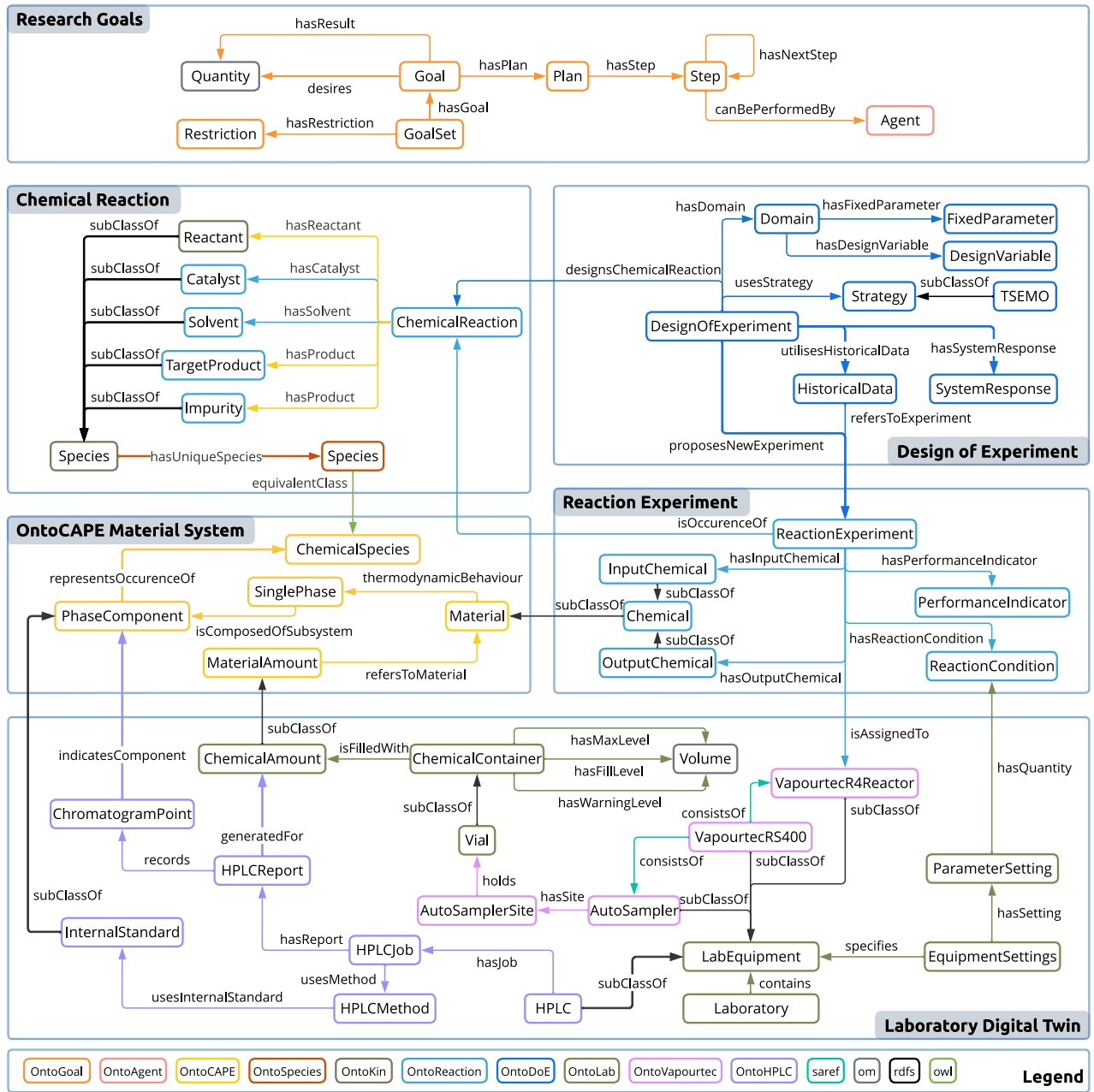

**Fig. 3 | A selection of concepts and relationships capturing different aspects in self-driving laboratories (SDLs).** The concepts are categorised based on their level of abstraction, spanning from high-level research goals to conceptual descriptions of chemical reactions and the mathematical expression of design of experiments, as well as the physical execution of reaction experiments and the laboratory digital twin. These concepts are interlinked with the OntoCAPE Material System, representing an effort to enhance interoperability with the community initiatives. Their namespaces correspond to the colour coding. For complete knowledge representation and namespace definitions see Supplementary Information section A.1.1.

OntoLab:containsUnidentifiedComponent for OntoLab:‐ChemicalAmount, a concept used for representing the concrete appearances of chemicals in the physical world. In terms of the collected reaction products, this representation is employed to signify the existence of (at least one) OntoHPLC:Chromato‐gramPoint in the OntoHPLC:HPLCReport that is designated OntoHPLC:unidentified. A more comprehensive representation of impurities can be achieved in conjunction with concentration-related concepts, such as OntoCAPE:Molarity, which we shall incorporate in future work. For concrete examples of ontology instantiation see Supplementary Information section A.1.

## Goal-driven knowledge dynamics

Figure 5 presents a high-level overview of the goal-driven evolution of the knowledge graph during closed-loop optimisation. The dynamicity of the knowledge graph is enabled by the presence of software agents that realise each component of the distributed architecture and facilitate the flow of information within the graph. The process begins with the goal derivation stage where the scientist initiates a goal request. The Reaction Optimisation Goal (ROG) Agent translates this request into a machine-readable statement that captures the scientist's intention. To accommodate all objectives, a goal set is formulated that considers each objective as a reward function for the agents'

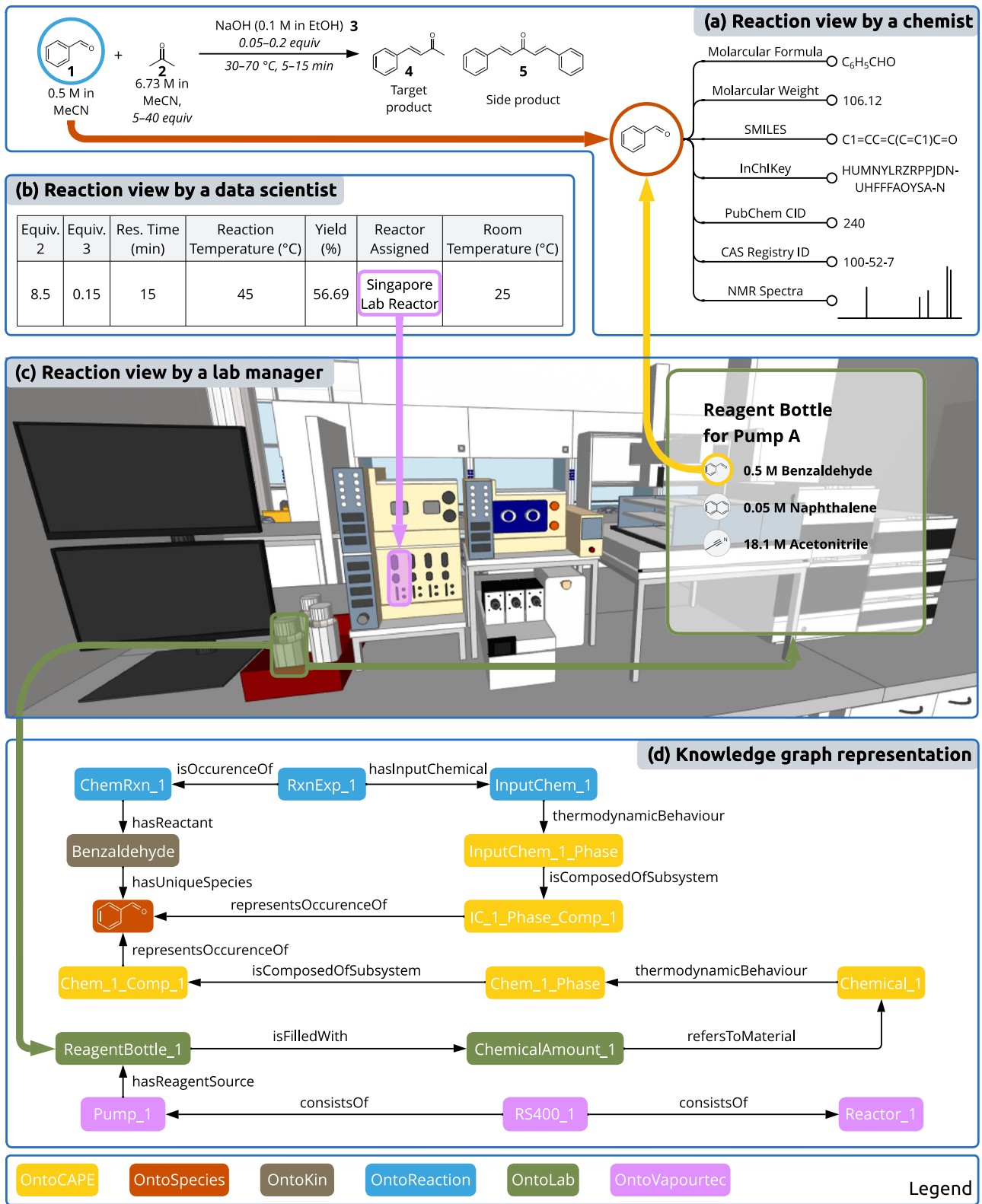

**Fig. 4 | A snapshot of reaction views from different perspectives. a** A chemist view of a reaction is based on the chemical structures. **b** A data scientist view of a reaction is based on the experiment conditions and resulting performance indicators. **c** A lab manager view of a reaction is based on hardware status and chemical availability. **d** The knowledge graph representation puts chemical informatics into context, allowing for queries and answers across these varied layers of abstraction (views). The colour coding corresponds to the ontological expression.

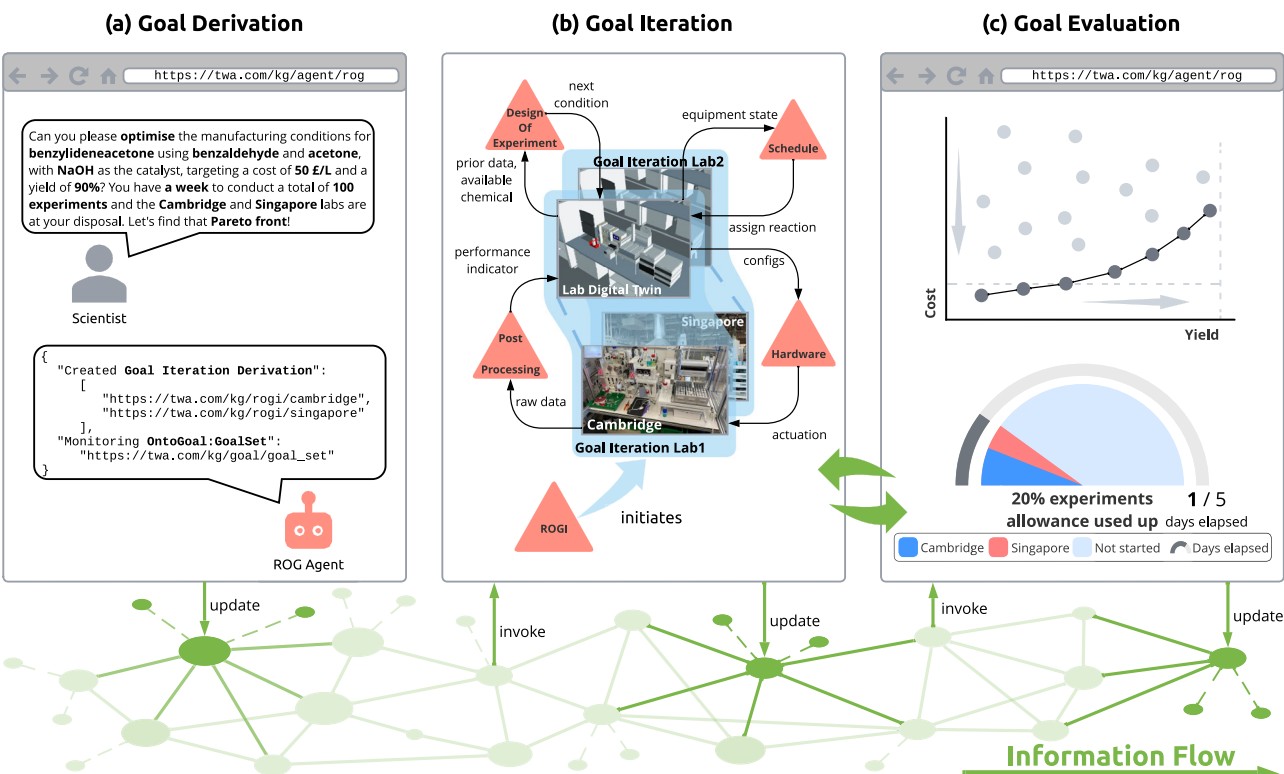

**Fig. 5 | Autonomous workflow triggered in response to goal requests from scientists as information travels within the knowledge graph. a** The Reaction Optimisation Goal (ROG) Agent translates the specification of a scientist into a machine-readable statement and instantiates it into the knowledge graph. **b** The Reaction Optimisation Goal Iteration (ROGI) Agent initiates the design-make-test-analyse cycle, during which other agents query information from the digital twin and actuate the hardware. **c** The progress of goal pursuit is assessed after each iteration, determining whether to proceed to the next cycle. Steps (**b**) and (**c**) are iterated until either goals are achieved or resources are depleted.

operations. For each participating laboratory, a `Goal Iteration Derivation` instance is created using the derived information framework[42] and requested for execution by the Reaction Optimisation Goal Iteration (ROGI) Agent.

The goal iteration stage plays a central role in the evolution of the dynamic knowledge graph. It involves the ROGI Agent initiating the flow of information among the participating agents towards achieving the goals. This process begins with the ROGI Agent creating tasks for the corresponding agents according to the DMTA cycle, including the DoE Agent, Schedule Agent, and Post-Processing Agent. The DoE Agent perceives the knowledge graph to retrieve prior data and chemical stock available for experiments and then proposes a new experiment. The Schedule Agent evaluates the hardware available in the specified laboratory according to the proposed conditions and subsequently selects the most appropriate hardware to execute the experiment. This is accomplished by generating tasks for the agents responsible for managing the selected digital twin. These agents actuate the equipment to perform reaction and characterisation in the physical world. When the HPLC report is generated, the Post-Processing Agent analyses the chromatogram data to calculate the objectives.

During the third stage, the ROG Agent utilises the obtained results to determine whether the next iteration should be pursued. To do so, it checks if the Pareto front of the multi-objective fulfils the pre-defined goals and if the resources are still available. The reaction experiment performed in the current iteration then becomes historical data, serving as input for the succeeding round of the `Goal Iteration Derivation` across all participating SDLs. Afterwards, a new request will be made to the ROGI Agent to start a new iteration, forming a self-evolving feedback loop.

To ensure correct data dependencies and the order of task execution, we employed the derived information framework[42] to manage the iterative workflow. We implemented each software agent using the derivation agent template provided by the framework. Once deployed, these agents autonomously update the knowledge graph to actively reflect and influence the state of the world.

This approach enables flexibility and extensibility in the system. As the digital twin of each lab is represented as a node in the knowledge graph, new hardware can be added or removed during the optimisation campaign by simply modifying the list of participating laboratories. The experimental allowance can also be updated when more chemicals become available. The system also supports data sharing across organisations at the very moment the data are generated. Details on the internal logic and technical aspects of the agents in the knowledge graph implementation are available in the Supplementary Information section A.2.

### Collaborative closed-loop optimisation

To demonstrate the scalability and modularity, the knowledge graph approach was applied to a real-time collaborative closed-loop optimisation distributed over two SDLs in Cambridge and Singapore. The objectives selected are run material cost and yield that were sampled for a search space of molar equivalents (relative to benzaldehyde **1**) of acetone **2**, NaOH **3**, residence time and reaction temperature. The research goals and restrictions were populated in the knowledge graph via a web front end. As no prior experimental data was provided, the agents start experiments with random conditions and gradually update their beliefs using TSEMO algorithm[56]. Before running the optimisation, two labs were verified to produce consistent results for

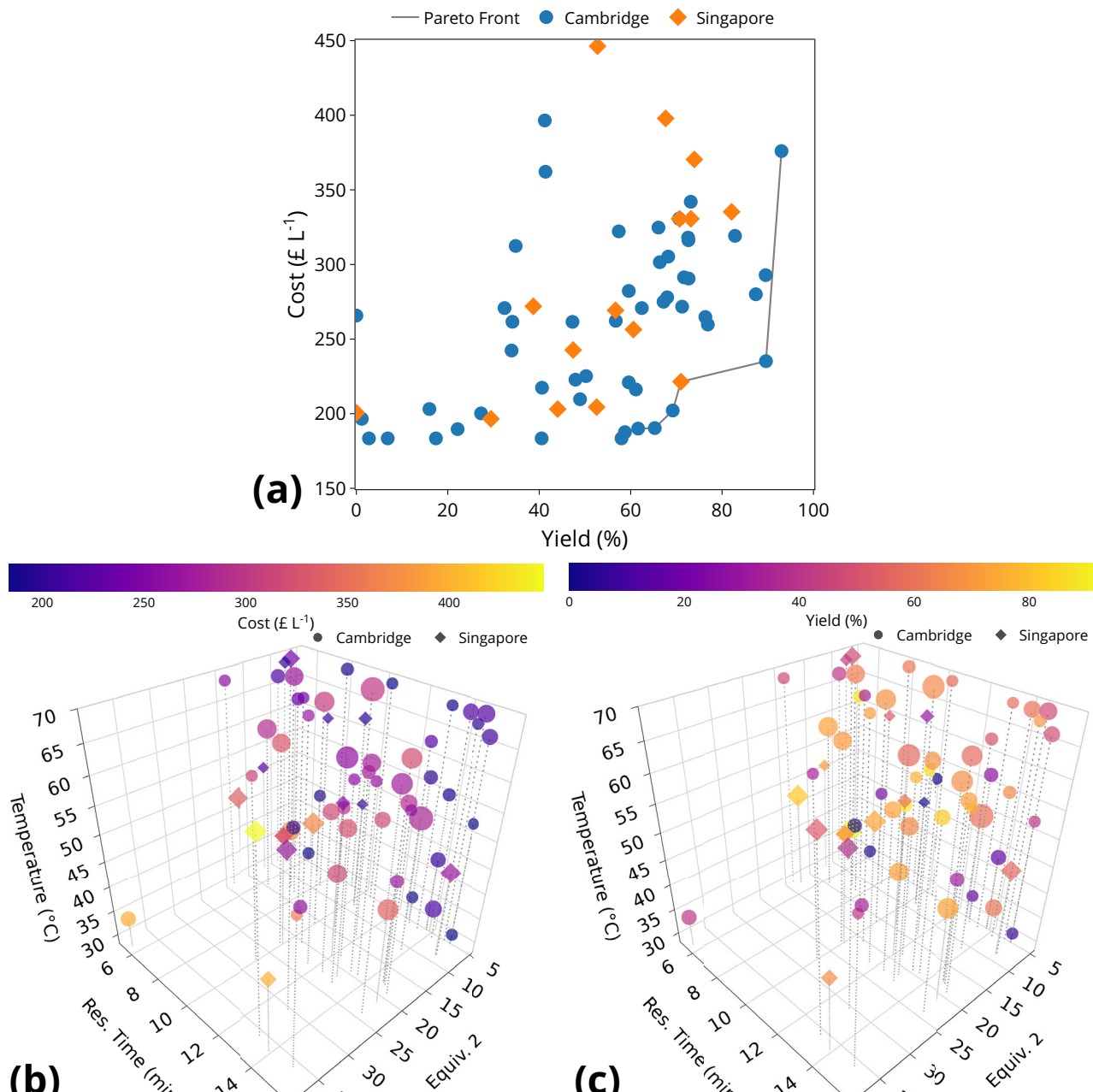

**Fig. 6 | Objectives and design variables of experiments conducted in the closed-loop optimisation campaign in distributed self-driving laboratories (SDLs).** Each dot refers to a single run. The animation of the optimisation progress is available in Supplementary Movie 1. Interactive versions of 3D plots are available in Supplementary Movies 2 and 3 for cost and yield objectives, respectively. Source data are provided as a Source Data file. **a** Pareto front plot of the yield and cost objectives for the aldol condensation reaction collaboratively optimised by two distributed SDLs. **b** Three-dimensional plot of the four sampled design variables colour coded for run material cost during the closed-loop optimisation. The size of the dots denotes the molar equivalents of 3 in each run. **c** Three-dimensional plot of the four sampled design variables colour coded for yield during the closed-loop optimisation. The size of the dots denotes the molar equivalents of 3 in each run.

two control conditions, in line with the practice of Shields et al.[57]. For experimental details see Supplementary Information section A.3.

Figure 6a presents the cost-yield objectives consisting of 65 data points collected during the self-optimisation. Throughout the operation, two SDLs share the results with each other when proposing new experimental conditions. The real-time collaboration demonstrated faster advances in the Pareto front with the highest yield of 93%. The chemicals used in this study were obtained from different vendors compared to ref. 51, the cost is therefore not directly comparable due to different prices. Although not considered in the optimisation, the environment factor and space-time yield were found to be highly

correlated to the yield objective. The best values obtained are 26.17 and 258.175 g L$^{-1}$ h$^{-1}$ when scaled to the same benzaldehyde injection volume (5 mL), both outperformed the previous study[51].

Figure 6b, c illustrate the influence of the continuous variables on the cost and yield objectives, with their interactive versions as Supplementary Movie 2 and 3, respectively. The cost is calculated to count for the molar amount of input chemicals sourced from the pumps for the reaction. Therefore, it increases linearly with the molar equivalents of the starting materials. Similarly as identified by ref. 51, reaction temperature has a positive correlation with the yield of reaction, whereas the residence time shows a poor correlation.

Upon examination of the molar equivalent of acetone **2**, it can be observed that its further increase after 30 results in a reduction in yield. This decrease can be attributed to the formation of more side product **5** and other further condensation products of acetone and benzaldehyde.

Notably, the Singapore setup encountered an HPLC failure after running for approximately 10 h. This caused peak shifting of the internal standard which resulted in a wrongly identified peak that gives more than 3500% yield. This point is considered abnormal by the agents and therefore not utilised in the following DoE. An email notification was sent to the developer for maintenance which took the hardware out of the campaign. The asynchronous and distributed design enabled the Cambridge side to further advance the Pareto front for the cost-yield trade-offs. It is also notable that the product peak was missed for one run at the Cambridge side due to a small shift of the peak which gives a yield of 0%. This point was taken into consideration in the DoE, but fortunately, it did not affect the final Pareto front as the corrected yield is still Pareto-dominated. The optimisation campaign was stopped since no more significant improvement was observed in terms of hypervolume, and also due to requests for repurposing the equipment for other projects. The complete provenance records (knowledge graph triples) are provided as Supplementary Data, along with an interactive animation of the optimisation progress extracted from them as Supplementary Movie 1.

## Discussion

In this contribution, we presented a dynamic knowledge graph approach to realise a conceptual architecture for distributed SDLs. We developed ontologies to represent various aspects of chemical knowledge and hardware digital twins involved in a closed-loop optimisation campaign. By employing autonomous agents as executable knowledge components to update and restructure the knowledge graph, we have enabled collaborative management of data and material flow across SDLs. Our approach allows scientists to initiate the autonomous workflow by setting up a goal request, which triggers the flow of information through the knowledge graph as the experimentation workflow progresses.

As a proof-of-concept demonstration, we applied the system to an aldol condensation reaction using two setups across different parts of the globe. Despite the differences in configurations, the reaction data produced by both machines were interoperable owing to the layered knowledge abstraction. Throughout the experiment, the system recorded all data provenance as the knowledge graph evolved autonomously, providing opportunities for informed machine learning[58]. Our collaborative approach resulted in faster data generation and advanced the Pareto front while exhibiting resilience to hardware failure.

The implementation of this work has provided valuable insights and identified areas for future improvement in the realm of dynamic knowledge graph systems. In terms of orchestration, it is crucial for the system to be robust to network disruption since it is distributed over the internet. We have implemented measures to ensure that agents deployed in the lab can handle internet cut-offs and resume operations once back online. To minimise downtime during reconnection, future developments could provide on-demand, localised deployment of critical parts of the knowledge graph to sustain uninterrupted operation.

For efficient optimisation and data quality, it is critical to have control conditions in place when adding new setups to the network, and only those generated results within the tolerance should be approved. Complex reactions with high-dimensional domains may not be sufficiently evaluated using only two control conditions. This highlights the persisting challenges in maintaining data quality and opens avenues for incorporating strategic cross-workflow validation experiments.

To increase the system's robustness against software and hardware malfunctions, regular backups of all data in the central knowledge graph should be implemented. Hardware failures during the self-optimisation campaign, which resulted in abnormal data points, also revealed an unresolved issue in automated quality control monitoring. This accentuates the need for a practical solution to bridge the interim technology gap, such as implementing a human-in-the-loop strategy for the effective monitoring of unexpected experimental results.

Further development could also be made to federate the SDLs, where each lab hosts its data and digital twins locally and only exposes its capabilities in the central registry (a "yellow page") without revealing confidential information. An authentication and authorisation mechanism should be added to control access to the equipment and grant permission for federated learning.

When reflecting on the vision of distributed SDLs, our approach exhibits both commonalities and distinctions when compared to contemporary designs. Table 1 summarises the key design features, to the best of our knowledge, as they relate to the three major challenges, with the first challenge further divided into the abstraction of resources and workflow coordination.

In terms of resource abstraction, all approaches (including the one presented in this work) employ a modular design that considers hardware limitations in granularity. This modularity is key for a seamless integration of new resources into a plug-and-play system. However, the way resources are exposed to the coordinator varies and this significantly impacts the orchestration of workflows across laboratories. This applies to both workflow template encoding and its actual execution. The dynamic knowledge graph approach uses agents acting as lab resource wrappers with knowledge graph access. Agents can register for jobs and proactively execute tasks assigned to the digital twin of the resources they manage. This approach is preferable compared to the practices in the remote procedure call paradigm, where lab resources are made accessible as web servers. Based on our experience, it can raise concerns among IT staff when exposing resources across university or company firewalls. Similar to agents, our approach encodes the workflow in the knowledge graph with each step overseen by an agent. Compared to encoding workflows as a sequence of function calls in scripting languages (such as Python), where execution may struggle with asynchronous workflows evolving during optimisation, our approach allows for real-time workflow assembly and modification. For a detailed technical discussion, interested readers can refer to the derived information framework[42].

The integration of data serialisation and storage within workflow aims to ease community adoption. As seen in Table 1, practices range from transmitting diverse file formats to enforcing a unified data representation. Starting with ad hoc extraction-transformation-loading tools for new devices prototyping is practical and minimally disruptive when upgrading a single lab. However, we find this approach less effective for scaling up to a large network of SDLs[32]. This limitation is the driving force behind the development of the dynamic knowledge graph approach, despite the initial cost required for creating ontologies that capture a collective understanding of the field. Our design delegates the responsibility of digesting and translating ontologies into the requisite language and file formats to autonomous agents. Compared to adopting a central coordinator to handle data transfer and format translation, our approach emphasises information propagation within a unified data layer, obviating the need for peer-to-peer data transfer and alleviating network congestion. Drawing an analogy to self-driving cars, once the "driving rules" (ontologies) are learned, SDLs are granted permission to drive on the "road" (information flow). Compared to traditional relational databases used in other studies, where schema modification can be challenging, the open-world assumption inherent in the dynamic knowledge graph enhances its extensibility. Organising concepts and relationships within a knowledge graph is also more intuitive than

**Table 1 | Comparison between contemporary designs and this work towards the realisation of distributed self-driving laboratories (SDLs)**

| Reference | Resource abstraction | Workflow orchestration | Data serialisation and storage | Experimental provenance |
|---|---|---|---|---|
| SiLA[28] | Hardware functions are abstracted as SiLA Features following a micro-service architecture with their behaviour described as a state machine. | A sequence of function calls using gRPC and HTTP/2 protocols. | AnIML[28] is employed as a file-less medium for bidirectional analytical data transmission between laboratory information management systems and chromatography data systems. | Device metadata collected during measurements are stored in XML files. |
| ChemOS[76] | Both software and hardware are abstracted as SiLA servers. Time-consuming computational jobs are managed by AiiDA[77] on SLURM[41]. | The central coordinator executes a sequence of Python function calls. The coordinator is also responsible for creating job files in the format required by each hardware/software. | Data are streamed between the central coordinator and each device in diverse file formats, e.g., pickle object, JSON, and CSV. The storage is done in an internal database with a schema consisting of device-agnostic and device-specific tables. | Job execution logs are stored with timestamps in the database table corresponding to each device. |
| ESCALATE[24] | The Django framework is used to abstract resources as REST API endpoints. | A sequence of steps encoded for "ExperimentTemplate" and accessible via REST API endpoints. | Data are stored in a PostgreSQL database and served via Django REST API[78] endpoints that serialise the data into JSON format for web transfer and inspection. | Metadata associated with the execution of workflow steps are stored with experiment instances in the relational database. |
| HALEO[25] | Device and their functions ("actions") are represented as hierarchical and asynchronous FastAPI[79] web servers. | A central coordinator executes a sequence of API calls (wrapped as Python functions). | Data are recorded as "groups" and "datasets" in the HDF5 file format and deposited into institutional repositories. | Metadata of "actions" are recorded in the same HDF5 file as the experimental measurements. |
| χDL[26,80] | Hardware is categorised/abstracted based on the unit operations in chemical reactions that it can execute. | The sequence of synthesis steps is expressed in XML format, which is later compiled into machine-actionable instructions in a Python script. | The analysis reports are kept in their native file format and bundled with the synthesis description in a PostgreSQL database. | Hardware instructions, the actual performed actions, and other metadata are stored with experiment instances in the relational database. |
| This work | Hardware is virtualised as a digital twin in a knowledge graph, where its control interface, akin to software resources, is wrapped using the derivation agent template. | DMTA cycles are expressed as directed acyclic graphs of "derivations" in the knowledge graph each referring to a knowledge graph step managed by the respective software agent. | Data are expressed in ontological format (triples) wherever possible. Files (e.g., CSV and XLS) are stored on a file server. Their ontological translation and pointers to the file server location are stored in the triple store. | The inputs/outputs annotation of each step in the workflow is recorded by the derived information framework as triples. The detailed operation timing is not recorded owing to API limitations in obtaining information at this level of granularity. |

traditional tabular structures. However, this flexibility may come at the cost of performance issues when handling extensive data volumes, especially when dealing with data on the scale of ORD. To counter this, technologies such as ontology-based data access[59] can create a virtual knowledge graph from relational databases, combining the strengths of both approaches.

Our approach to experimental provenance differs from others due to hardware constraints. It focuses less on exact operation timing, such as robotic arm motions, and more on capturing inputs and outputs within DMTA cycles. This facilitates high-level analysis, enabling answering questions like "which experiments from lab A informed the DoE study for a specific reaction in lab B". This capability has been effectively demonstrated in the interactive Pareto progress animation provided in Supplementary Movie 1. However, for a deeper understanding of epistemic uncertainties associated with operations in complex reactions, it is imperative to expand the ontologies for a more granular abstraction of the experimental procedures. A potential expansion in this regard could involve the ontologisation of χDL.

Looking forward, achieving a globally collaborative research network requires collective efforts. As the knowledge graph aims to reflect a communal understanding of the field, involving different stakeholders early on can accelerate collaboration and increase the chance of success. Specifically, there exists an opportunity for using knowledge graph technology as an integration hub for all aforementioned initiatives. Industrial partners are encouraged to work together and provide a unified API for interacting with their proprietary software and hardware interfaces. This can be facilitated by efforts such as OPC UA[60] and SiLA[28]. Recent studies have shown the successful exchange of HPLC methods between vendors in the Chromatography Data System (CDS), demonstrating the potential for the ontology-based approach[61]. Collaboration between scientists and industry is also important at various stages of research and development[62].

Overall, we believe the dynamic knowledge graph approach demonstrated in this work provides the first evidence of its potential to establish a network of globally distributed SDLs. Although we focus on flow chemistry in this study, the principles are generic. The same approach can be applied to DMTA cycles for other domains should relevant ontologies and agents be made available, for example, to support research in deep space[18].

## Methods

### The World Avatar knowledge graph

This work follows the best practices in the World Avatar project. All ontologies and agents are version-controlled on GitHub. We provide our thought process during the development below. The same principles can be followed for self-optimisation applications in other domains.

**Ontology development.** Developing ontologies is often an iterative process and it is not a goal in and of itself[63]. As suggested in[32,36,37], we follow the steps from specifying target deliverables to conceptualising relevant concepts and finally implementing codes for queries. Aimed at capturing data and material flow in distributed SDLs, the relevant concepts range from the reaction experiment to the hardware employed to conduct it. In the World Avatar, ontologies are typically developed to be digested by software agents which mimic the human way of conducting different tasks[64]. Therefore, the development draws inspiration from relevant software tools[51,65,66] and existing reaction database schemas[48,49]. Views of the domain experts[67–69] are also consulted to better align with the communal understanding of the subject. During iterations, competency questions are used to test if the ontologies meet case study requirements. The answers to these questions are provided in the form of SPARQL queries that are executed by the agents during their operations. Another essential aspect to consider is data instantiation, where we adopted `pydantic` to

simplify the querying and processing of data from the knowledge graph. Overall, the ontology development process starts as easily as drawing concepts and their relationships on a whiteboard and then gradually materialising them in code.

**Agent development.** Following the development of ontologies, agents are defined as executables that process inputs and generate outputs. Their I/O signatures are represented following OntoAgent[70]. At the implementation level, all agents inherit the `DerivationAgent` template in Python provided by the derived information framework[42]. Specifically, agents utilise the asynchronous communication mode when interacting with the knowledge graph as conducting experiments is inherently a time-consuming process. Each of the agents monitors the jobs assigned to itself and records the progress of execution in the knowledge graph. The derived information framework does most of the work behind the scenes, leaving the developer with the only task of implementing each agent's internal logic. As agents modify the knowledge graph and subsequently actuate the real world autonomously once active, it is important to make sure they behave as expected. In this regard, unit and integration tests are provided to help with responsible development. For instance, the integration tests in folder RxnOptGoalAgent/tests simulate the behaviour of distributed SDLs to verify that the data flows are as expected upon goal request from scientists. Detailed descriptions of tests for each agent can be found in section A.2 of the Supplementary Information.

**Distributed deployment.** Taking inspiration from remote control practices in lab automation[71–73], the knowledge graph is designed to span across the internet. It follows deployment practices commonly used by cloud-native applications and is implemented through docker containers. The triplestore and file server containing the knowledge statements are deployed at internet-resolvable locations. Depending on capabilities, agents are located at different host machines. Those who monitor and control the hardware are deployed in the corresponding laboratory for security reasons. They transmit data collected from the hardware to the knowledge graph and in reverse configure and actuate the equipment when a new experiment arises. At start-up, agents register their OntoAgent instances in the knowledge graph, then act autonomously should tasks be assigned to them. Altogether, these agents form a distributed network that facilitates the transfer of information within the knowledge graph and bridges cyberspace and the physical world.

**Flow chemistry platforms**
This work connects two similar automated flow chemistry platforms located in Cambridge and Singapore. The method of sourcing input chemicals differs, with a liquid handler employed in Cambridge and reagent bottles utilised in Singapore. We provide below brief descriptions of the experimental setup. All chemicals were used as received.

**Cambridge lab.** On the Cambridge side, the experimental setup consists of two Vapourtec R2 pump modules, one Vapourtec R4 reactor module, one Gilson GX-271 liquid handler, one four-way VICI switching valve (CI4W.06/.5 injector), and Shimadzu CBM-20A HPLC analytical equipment equipped with Eclipse XDB-C18 column (Agilent part number: 993967-902). To initiate the reaction, the liquid handler dispenses a 2 mL solution of 0.5 M benzaldehyde **1** dissolved in acetonitrile (with 0.06 M biphenyl as an internal standard) into the sample loop of pump A. Acetone **2** (50% v/v in acetonitrile) and 0.1 M NaOH **3** in ethanol are similarly loaded into sample loops for pump B and C. After being transferred by the switching valve, the product (benzylideneacetone **4**) is analysed using online HPLC. The HPLC analysis lasts 17 min, with a mobile phase consisting of an 80:20 (v/v) binary mixture of water and

acetonitrile running at a rate of $2\,mL\,min^{-1}$. All compounds are detected at an absorption wavelength of 254 nm.

**Singapore lab.** On the Singapore side, the experimental setup consists of two Vapourtec R2 pump modules, one Vapourtec R4 reactor module, one 6-port 2-position VICI switch valve equipped with 60 nL sampling rotor, and an Agilent 1260 Infinity II system equipped with a G1311B quaternary pump, Eclipse XDB-C18 column (Agilent product number: 961967-302), and G1314F variable wavelength detector (VWD). The input chemical for the reaction is sourced from three reagent bottles that are directly attached to the Vapourtec pumps: pump A contains 0.5 M benzaldehyde **1** in acetonitrile (with 0.05 M naphthalene as an internal standard), pump B contains 6.73 M acetone **2** in acetonitrile (50% v/v in acetonitrile), and pump C contains 0.1 M NaOH **3** in ethanol. The following HPLC quaternary pump method for online HPLC is used: the initial mobile phase was a 5:95 (v/v) binary mixture of acetonitrile and water flowing at $0.2\,mL\,min^{-1}$. Immediately after sample injection, the flow rate and ratio of acetonitrile to water were steadily changed to $1\,mL\,min^{-1}$ and 95:5 (v/v) during the first 5 min. At a flow rate of $1\,mL\,min^{-1}$, the binary mixture ratio is then returned to 5:95 (v/v) acetonitrile:water over 1.5 min in a linear gradient. This binary mixture ratio is held constant at $1\,mL\,min^{-1}$ for the next 1.5 min, after which the analysis is complete (after a total of 8 min), and the method returns to a flow rate of $0.2\,mL\,min^{-1}$. The VWD wavelength was changed over the 8 min analysis time as follows: the absorption wavelength is 248 nm for the initial 6.05 min and then switched to 228 nm until the end of acquisition.

**Reporting summary**
Further information on research design is available in the Nature Portfolio Reporting Summary linked to this article.

## Data availability
Research data generated in this study has been deposited in the University of Cambridge data repository under accession code https://doi.org/10.17863/CAM.97058[74]. Source data are provided with this paper. The tabular format of relevant experimental results that were displayed in Fig. 6 is provided in the Source Data XLSX file. Source data are provided with this paper.

## Code availability
All the codes developed are publicly available on The World Avatar GitHub repository https://github.com/cambridge-cares/TheWorldAvatar or the Zenodo repository at https://doi.org/10.5281/zenodo.10151236[75]. The docker images of agents are available at GitHub's public registry located at `ghcr.io/cambridge-cares/`: doe_agent:1.2.0, vapourtec_schedule_agent:1.2.0, vapourtec_agent:1.2.0, hplc_agent:1.2.0, hplc_postpro_agent:1.2.0, rxn_opt_goal_iter_agent:1.2.0, and rxn_opt_goal_agent:1.0.0. The deployment instructions can be found in folder TheWorldAvatar/Deploy/pips.

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

## Acknowledgements

This research was supported by the National Research Foundation, Prime Minister's Office, Singapore, under its Campus for Research Excellence and Technological Enterprise (CREATE) programme, and Pharma Innovation Platform Singapore (PIPS) via grant to CARES Ltd "Data2Knowledge, C12". This project was cofunded by European Regional Development Fund via the project "Innovation Centre in Digital Molecular Technologies", UKRI via project EP/S024220/1 "EPSRC Centre for Doctoral Training in Automated Chemical Synthesis Enabled by Digital Molecular Technologies". Part of this work was also supported by Towards Turing 2.0 under the EPSRC Grant EP/W037211/1. The authors thank Dr. Andrew C. Breeson for his helpful suggestions on graphical design. J.B. acknowledges financial support provided by CSC Cambridge International Scholarship from Cambridge Trust and China Scholarship Council. C.J.T. is a Sustaining Innovation Postdoctoral Research Associate at Astex Pharmaceuticals and thanks Astex Pharmaceuticals for funding, as well as his Astex colleagues Chris Johnson, Rachel Grainger, Mark Wade, Gianni Chessari, and David Rees for their support. S.D.R. acknowledges financial support from Fitzwilliam College, Cambridge, and the Cambridge Trust. M.K. gratefully acknowledges the support of the Alexander von Humboldt Foundation. For the purpose of open access, the author has applied a Creative Commons Attribution (CC BY) licence to any Author Accepted Manuscript version arising.

## Author contributions

M.K., A.A.L., J.B. and S.M. conceived the project. J.B., S.M. and M.K. designed the ontological representation and agent workflow. J.B. implemented the ontologies/agents and deployed the knowledge graph under the advisement of S.M. and K.F.L. The chemistry and HPLC method were developed by C.J.T. (Cambridge) and D.K. (Singapore). D.K. and C.J.T. validated the calculation of objective functions. The self-optimisation campaign involving two labs was set up by C.J.T. (hardware and chemicals in Cambridge), D.K. (hardware and chemicals in Singapore) and J.B. (software on both sides and goal request). M.K. and A.A.L. acquired funding and administrated the project. J.B. draughted the body of this manuscript and SI with inputs from S.M., J.A., S.D.R. and C.J.T. All authors provided feedback on the manuscript.

## Competing interests

The authors declare no competing interests.
