## [Peer Review File · Nature Communications]

From Platform to Knowledge Graph: Distributed Self-Driving LaboratoriesREVIEWER COMMENTS

Reviewer #1 (Remarks to the Author):

In the rapidly advancing field of automation for accelerated scientific discovery, there have been many examples of automated experiments and algorithms for computational guidance. There have also been a few nascent discussions of ontologies and knowledge graphs, which are underappreciated topics brought to the center of attention in the present submission. The article by Bai et al comprises an enormous step forward in formalizing AI-driven science via dynamic knowledge graphs that coordinate research across automated workflows. The work is remarkable in its development and demonstration of machine encodings for research goals, search strategies, and scientific questions. The paper presents not only the vision for the future of AI assisted research, but also an incredible amount of progress towards realizing that vision. This is the most original and important paper I have seen this year in the field of accelerated science, and I strongly support its publication after a few items are addressed.

Overall the discussion of prior literature is excellent. I am not some publications that have come out since the submission was prepared, which the authors may want to consider:

1. A recent perspective article provides excellent motivation for the present work. I think this can be cited in general and specifically with respect to the role of interconnected labs in automating characterization of epistemic uncertainty:

Ren, Z.; et al. Autonomous Experiments Using Active Learning and AI. Nat Rev Mater 2023, 1–2. <https://doi.org/10.1038/s41578-023-00588-4>.

2. Regarding data provenance, I think it's fine to cite "Mitchell et al. [28]" but you may also want to consider a recent example in materials/chemistry:

Statt, M. J.; et al The Materials Experiment Knowledge Graph. Digital Discovery 2023. <https://doi.org/10.1039/D3DD00067B>.

Here are a few items that may require some attention:

3. On p. 13 the authors describe experiments that "validate the reproducibility across the two setups" using two control conditions. Two conditions will never be sufficient to characterize reproducibility across the entire domain, so the authors may want to qualify their statement a bit and note the challenge and future opportunity to incorporate strategic selections of cross-workflow validation experiments.

4. On p. 13 there are also some descriptions of HPLC failures and other data artifacts. It is my own opinion that quality control is the hardest thing to automate and a self-driving lab, and it appears the examples in the present work support that opinion. The authors they want to highlight that this is an unsolved problem and that in the interim it may be best to adopt a human-in-the-loop strategy to monitor quality control, especially when a surprising experimental result is obtained.

5. Regarding "unit and integration tests are provided to help with responsible development." This is a subtle but important topic and doesn't necessarily need a lot of discussion, but it would be helpful if the authors could describe in a little more detail or point to specific examples in their repository.

Reviewer #3 (Remarks to the Author):

In this work, the authors present an ontological system that abstracts many layers of experimental chemistry, from molecules all the way up to optimization campaigns. They explain its function through a demonstration of delocalized, autonomous Pareto optimization. In this example, they optimize the cost and yield of a toy reaction (aldol condensation of benzaldehyde and acetone).

The key point of the authors' work is the abstraction framework that they have developed rather than the delocalized nature or the experiments themselves. Developing a universal abstraction for chemistry is really challenging since it requires understanding the experiments not only at a conceptual level, but also the practical issues that often arise. This is a thorough contribution to this goal. However, I think there are a number of ways in which the authors could reframe their work to make it easier for the reader to understand and appreciate.

Main question: Who is this aimed at? Since the authors are re-using existing ontological frameworks, I don't think the goal is to convince experts in that field. Rather, the language and structure of the text could be reformulated to target a broader audience of chemists rather than only those that are already very familiar with SDLs. I also disagree with the use of terms such as "knowledge engineer", "digital scientists", "AI scientists", etc. These terms are vague, unnecessarily jargony and obscure the fact that this work is expanding upon and refining other works and concepts that the authors cite.

The authors should also spend more time comparing their work to other systems that already exist, many of which they mention (ChemOS, ESCALATE, HELAO, etc). Namely, this framework contains many similar elements to HELAO, XDL, SiLA. Especially the latter! This work would benefit from a more thorough discussion of the comparative strengths/weaknesses of the authors' system compared to the others.

I would like to commend the authors for the following:

- Well organized GitHub and available documentation
- Considering hybrid labs, not only fully-automated
- Re-using existing frameworks (OntoCAPE, SAREF) rather than creating an entirely new system from scratch
- Tying the material to PubChem data

Shortcomings on the chemistry side:

- ChemicalReaction only accounts for what in the reaction, but not the order/timing of operations, which is known to be a crucial factor in some reactions
- The ontology assumes that the material is pure, which is rarely the case
- The synthesis isn't particularly interesting

Minor points/recommendations:

- Fig 4: It's unclear why a "3D digital twin" is necessary and why a lab manager would need to inspect it
- Fig 6b: 3D plots are very difficult to interpret. I suggest that the authors switch to 2D plots with color and marker size as the 3rd and 4th dimensions
- The Cambridge data repository link leads to a "DOI not found" page

Reviewer #1

“ In the rapidly advancing field of automation for accelerated scientific discovery, there have been many examples of automated experiments and algorithms for computational guidance. There have also been a few nascent discussions of ontologies and knowledge graphs, which are under-appreciated topics brought to the center of attention in the present submission. The article by Bai et al comprises an enormous step forward in formalizing AI-driven science via dynamic knowledge graphs that coordinate research across automated workflows. The work is remarkable in its development and demonstration of machine encodings for research goals, search strategies, and scientific questions. The paper presents not only the vision for the future of AI assisted research, but also an incredible amount of progress towards realizing that vision. This is the most original and important paper I have seen this year in the field of accelerated science, and I strongly support its publication after a few items are addressed. ”

We highly appreciate the reviewer’s recognition of our work as original and important to the field. We have addressed all the below points and we believe the manuscript has benefited significantly from them.

“ Overall the discussion of prior literature is excellent. I am not sure some publications that have come out since the submission was prepared, which the authors may want to consider:

1. A recent perspective article provides excellent motivation for the present work. I think this can be cited in general and specifically with respect to the role of interconnected labs in automating characterization of epistemic uncertainty: Ren, Z.; et al. Autonomous Experiments Using Active Learning and AI. Nat Rev Mater 2023, 1–2. <https://doi.org/10.1038/s41578-023-00588-4>.

2. Regarding data provenance, I think it’s fine to cite “Mitchell et al. [28]” but you may also want to consider a recent example in materials/chemistry: Statt, M. J.; et al The Materials Experiment Knowledge Graph. Digital Discovery 2023. <https://doi.org/10.1039/D3DD00067B>.
”

We thank the reviewer for stating these studies. We have added them accordingly in the revised manuscript.

“ Here are a few items that may require some attention:

3. On p. 13 the authors describe experiments that “validate the reproducibility across the two setups” using two control conditions. Two conditions will never be sufficient to characterize reproducibility across the entire domain, so the authors may want to qualify their statement a bit and note the challenge and future opportunity to incorporate strategic selections of cross-workflow validation experiments. ”

We acknowledge the reviewer’s comments and we have revised the sentence to clarify that the two laboratories produced consistent results for the two control conditions. In addition, we note this is similar to the approach adopted by Shields, B.J., Stevens, J., Li, J. *et al.* Bayesian reaction optimization as a tool for chemical synthesis. *Nature* 2021, 590, 89–96 doi:10.1038/s41586-021-03213-y, and we have added this to the manuscript. We have also added to the discussion in the manuscript to note the challenge and opportunity to include cross-workflow validation experiments.

“ 4. On p. 13 there are also some descriptions of HPLC failures and other data artifacts. It is my own opinion that quality control is the hardest thing to automate and a self-driving lab, and it appears the examples in the present work support that opinion. The authors they want to highlight that this is an unsolved problem and that in the interim it may be best to adopt a human-in-the-loop strategy to monitor quality control, especially when a surprising experimental result is obtained. ”

We completely agree with the reviewer. We have highlighted this point in the discussion in the revised manuscript.

“ 5. Regarding “unit and integration tests are provided to help with responsible development.” This is a subtle but important topic and doesn’t necessarily need a lot of discussion, but it would be helpful if the authors could describe in a little more detail or point to specific examples in their repository. ”

We thank the reviewer for this suggestion. We have pointed to an example of integration tests in the methodology section and provided more details for each agent in the Supplementary Information of the revised manuscript.

Reviewer #3

“ In this work, the authors present an ontological system that abstracts many layers of experimental chemistry, from molecules all the way up to optimization campaigns. They explain its function through a demonstration of delocalized, autonomous Pareto optimization. In this example, they optimize the cost and yield of a toy reaction (aldol condensation of benzaldehyde and acetone).

The key point of the authors’ work is the abstraction framework that they have developed rather than the delocalized nature or the experiments themselves. Developing a universal abstraction for chemistry is really challenging since it requires understanding the experiments not only at a conceptual level, but also the practical issues that often arise. This is a thorough contribution to this goal. ”

We are grateful for the reviewer’s comment.

“ However, I think there are a number of ways in which the authors could reframe their work to make it easier for the reader to understand and appreciate.

Main question: Who is this aimed at? Since the authors are re-using existing ontological frameworks, I don’t think the goal is to convince experts in that field. Rather, the language and structure of the text could be reformulated to target a broader audience of chemists rather than only those that are already very familiar with SDLs. I also disagree with the use of terms such as “knowledge engineer”, “digital scientists”, “AI scientists”, etc. These terms are vague, unnecessarily jargony and obscure the fact that this work is expanding upon and refining other works and concepts that the authors cite. ”

This work addresses the research gaps identified in our previous review (Fig. 2 in Ref. 32, doi:10.1021/jacsau.1c00438) of the current semantic representations for chemical digitalisation. Specifically, we have created a set of connected ontologies (*i.e.*, OntoGoal, OntoReaction, OntoDoE, OntoLab, OntoVapourtec, OntoHPLC) capturing different levels of abstraction in scientific research, *i.e.*, from research goals to the species level, while making efforts to ensure compatibility with other existing ontologies by re-using some of their concepts.

In response to the main question about our target audience, the primary objective of this manuscript is to showcase our developed technological solution through a practical application aimed at those interested in building and running self-driving laboratories (SDLs). We thank the reviewer for the suggestion to make it more accessible to reach a broader audience, especially those who may be interested in benefitting from the approach, without necessarily building and running it themselves. To achieve this, we have reviewed the body of the manuscript to simplify the language and reduce the

use of jargon wherever possible. We have additionally sampled feedback from several readers and have worked to enhance the clarity of the manuscript in response to their comments. Our focus has been making the body of the paper remain accessible to a wide readership, whilst keeping the technical details in the Supplementary Information. Furthermore, we would like to emphasise that all of our codes are publicly available. We believe these resources will provide valuable support to those looking to apply our technology in their laboratory settings, thereby facilitating the transition from local research to a more collaborative global research paradigm.

“ The authors should also spend more time comparing their work to other systems that already exist, many of which they mention (ChemOS, ESCALATE, HELAO, etc). Namely, this framework contains many similar elements to HELAO, XDL, SiLA. Especially the latter! This work would benefit from a more thorough discussion of the comparative strengths/weaknesses of the authors’ system compared to the others. ”

We thank the reviewer for this suggestion. During the preparation of the original manuscript, we carefully reviewed each relevant study. We have now included a table summarising the key features of these existing studies to the best of our knowledge. In addition, we have provided detailed discussions comparing our work with these selected systems.

“ I would like to commend the authors for the following:

- *Well organized GitHub and available documentation*
- *Considering hybrid labs, not only fully-automated*
- *Re-using existing frameworks (OntoCAPE, SAREF) rather than creating an entirely new system from scratch*
- *Tying the material to PubChem data ”*

We are grateful to the reviewer for the acknowledgement.

“ Shortcomings on the chemistry side:

- *ChemicalReaction only accounts for what in the reaction, but not the order/timing of operations, which is known to be a crucial factor in some reactions ”*

The `OntoReaction:ChemicalReaction` class presents a conceptual description of a chemical reaction, focusing solely on the reaction participants. The order and timing of the operations relate to the concrete realisation of such a reaction in the physical world, specifically within the `OntoReaction:ReactionExperiment`. However, in practice, the level of control will be constrained by the hardware itself and the available application programming interfaces (APIs). The proof-of-concept involves a single-step reaction in a flow system, with examples of the significant timing-related factors being the pump flow rate and the VICI valve switch for HPLC injection.

The reviewer is correct in noting that we did not take into account the order and timing of operations in our reaction experiments in this work. This limitation is a result of the version of `FlowCommander` we are using (v1.12, software for the reactor), which operates at a high level of encapsulation. Specifically, it takes an input file with condition parameters and carries out all the operations by itself. Despite the detailed operations visible in the graphical user interface (GUI), there were no available APIs that would allow us to log or control operations at a finer granularity. The ontologies were therefore simplified as a practical solution.

We agree with the reviewer that these aspects are important and we are open to expanding the ontology with additional concepts to accommodate these use cases, provided that the hardware allows for low-level control. We have added a sentence in the discussion stating one way of improving this could be ontologising XDL.

“ - *The ontology assumes that the material is pure, which is rarely the case* ”

We thank the reviewer for the suggestion on the material purity issue. The ontologies we developed contain the necessary concepts and relationships to address this concern. We employ the concept `OntoLab:ChemicalAmount` to represent the concrete appearances of chemicals in the physical world, and we use the data property `OntoLab:containsUnidentifiedComponent` to indicate the presence of the unidentified components (impurities). When dealing with starting materials, we can represent the impurities in reactants in this way. In the case of the collected reaction products, the same representation can be employed to indicate the existence of (at least one) `OntoHPLC:ChromatogramPoint` in the `OntoHPLC:HPLCReport` that is designated `OntoHPLC:unidentified`.

For the specific case study, we represented the impurities in the collected reaction products as described above. However, we simplified the representation of purity in the starting materials since they were used as received – all these materials were procured with purities exceeding 99% for liquid chemicals and 97% for NaOH pellets, which we believe to be sufficient. A more comprehensive representation for impurities would be incorporating concentration-related concepts such as `OntoCAPE:Molarity`. We have updated the manuscript to make this explicit and we shall instantiate the complete representation in our future use cases.

“ - *The synthesis isn't particularly interesting* ”

We deliberately selected aldol reactions as a proof-of-concept for the framework as they are ubiquitous in chemistry and have been well-studied. This choice was made with the intent of ensuring that our work is easily accessible and resonates with a broad audience.

Following the submission of this manuscript, we have subsequently applied the dynamic knowledge graph approach to some more interesting reactions (manuscript in preparation) and we are actively working on broadening the ontological representation to accommodate more complex synthesis processes, such as multi-step reactions involving a sequence of reaction and separation operations. These will be featured in upcoming manuscripts with a stronger chemistry focus.

We have clarified this in the section “Contextualised reaction informatics” of the revised manuscript where we first introduced the selected chemistry.

“ *Minor points/recommendations:*

- *Fig 4: It's unclear why a “3D digital twin” is necessary and why a lab manager would need to inspect it* ”

We acknowledge this might be unclear to readers. To clarify, we use the dynamic knowledge graph to consolidate a wide range of information sources relevant to the management of a research laboratory. The 3D digital twin, constructed based on instantiation in the knowledge graph, provides lab managers with a centralised information hub to monitor lab status. We have clarified this in the revised manuscript.

“ - Fig 6b: 3D plots are very difficult to interpret. I suggest that the authors switch to 2D plots with color and marker size as the 3rd and 4th dimensions ”

We thank the reviewer for the suggestion for improving the figures. However, we believe that adopting 2D plots may introduce challenges in effectively conveying the dimensionalities of the data presented. Figure 6(b)/(c) encompass six dimensions, comprising four design variables (residence time, reaction temperature, molar equivalent ratio for acetone and NaOH), one location variable (Cambridge vs. Singapore lab), and one objective function (cost or yield). Therefore, we have opted for 3D plots to help illustrate the complexity.

To improve the interpretability of the 3D plots, we created their interactive versions as additional Supplementary Video 2/3 for Fig. 6(b)/(c) respectively. These interactive resources allow readers to rotate the canvas and zoom in to explore the data more thoroughly, with metadata displayed in a floating window when hovering the mouse over the data points. We believe that these modifications will provide a more accessible and informative presentation of our results.

“ - The Cambridge data repository link leads to a “DOI not found” page ”

The University of Cambridge’s DOI policy (<https://www.repository.cam.ac.uk/about/doi-policy>) is that datasets cannot be changed (*e.g.*, files added, removed or amended) after their approval for inclusion in the repository. Their recommendation is that datasets should only be finalised (and the DOI in the manuscript therefore only becoming live) after acceptance of the manuscript. We realise that this creates a frustrating chicken-and-egg situation. For peer-review purposes, all data that will eventually be accessible via the DOI is available in the supplementary materials submitted with this manuscript.

REVIEWERS' COMMENTS

Reviewer #1 (Remarks to the Author):

Thank you for addressing all items. I believe this is ready for publication.

Reviewer #3 (Remarks to the Author):

The authors have thoroughly and satisfactorily addressed the comments of both reviewers.